# Coupling Coordinated Development and Exploring Its Influencing Factors in Nanchang, China: From the Perspectives of Land Urbanization and Population Urbanization

**Tiangui Lv [1,2], Li Wang [1], Xinmin Zhang [2], Hualin Xie [2,*], Hua Lu [2] , Hongyi Li [1], Wangda Liu [1] and Yanwei Zhang [1]**

[1]   School of Tourism and Urban Management, Jiangxi University of Finance and Economics,
     Nanchang 330013, China; lvtiangui@jxufe.edu.cn (T.L.); wangli9989@163.com (L.W.);
     lihongyi1981@zju.edu.cn (H.L.); liuwangda966@163.com (W.L.); bsonyan@126.com (Y.Z.)
[2]   Institute of Ecological Civilization, Jiangxi University of Finance and Economics, Nanchang 330013, China;
     zhangxm1217@yahoo.com (X.Z.); luhua@jxufe.edu.cn (H.L.)
[*]   Correspondence: xiehualin@jxufe.edu.cn; Tel.: +86-791-8397-9115

**Abstract:** The coordination relationship between land urbanization and population urbanization is crucial for achieving sustainable development under economic transition. Moreover, the balance between land urbanization and population urbanization is essential to guarantee the urbanization process of an entire city. This paper empirically analyzes the interaction between land urbanization and population urbanization in Nanchang from 2002 to 2017 based on the coupling coordination model (CCM). The impacts of the coordination degree on coordinated development are quantified by multivariate linear regression (MLR). The results show the following: (1) The indices of land urbanization and population urbanization in Nanchang showed an upward trend, and therefore the coordination degree in Nanchang increased significantly from 2002 to 2017. (2) The coordinated development of urbanization underwent two stages: disordered and moderately disordered. (3) The urban population proportion and the supporting capability of agricultural production had a positive impact on coordinated development. Meanwhile, the results also show that per capita education expenditures and the per capita public green area had negative impacts on the coordination degree, while economic development and the urban industrial structure were positive contributors to the coordination degree. Finally, this paper proposes that policies should be formulated to achieve coordinated development of urbanization. It can be concluded that the results regarding coordinated development of urbanization can help decision makers formulate effective measures to achieve coordinated development in the future.

**Keywords:** coordinated development; land urbanization; population urbanization; coupling coordination model (CCM); multivariate linear regression (MLR); Nanchang

## 1. Introduction

Urbanization generally refers to both the process of population concentration in urban areas and the transformation of rural areas into urban areas [1]. Since the middle of the 20th century, the coordinated development of land urbanization and population urbanization has received considerable attention [2]. The reason is that many countries and regions have witnessed the challenges arising from unsustainable development at the expense of the urban growth boundary (UGB) [2,3], as well as the growth in human activities associated with rapid land exploitation. In particular, since the

reform and opening-up initiated in the late 1970s, urbanization in China has accelerated rapidly due to farmland conversion (i.e., farmland being converted to land used for construction, etc.) and population migration (i.e., the rural population gradually migrating to urban areas, etc.), which has greatly enhanced the quality of life and economic development [4]. Furthermore, it is estimated that the urbanization growth rate is expected to reach 60% by 2030 and 70% by 2050 [5]. In this context, the coordination of land urbanization and population urbanization has become a hot topic, which is important to achieve coordinated development of urbanization.

However, unbalanced coordination between land urbanization and population urbanization is likely to cause social inequalities, informal settlements, slums, land scarcity and ecosystem degradation, all of which have a negative impact on a city's sustainable development [6]. Meanwhile, it has been suggested that rapid urbanization often occurs at the expense of the loss of agricultural land to satisfy the demand for urban land. In particular, issues such as the sedimentation of watersheds, urban pollution and population crowding are exacerbated as urban land is overexploited. Coordination between land urbanization and population urbanization is essential to efforts to cope with increasing urbanization and wealthier populations, for instance, the increasing amount of population migration from rural areas to urban areas. In other words, rapid urbanization has negative impacts on society in China, e.g., a soaring population, uncontrolled land use, a decreasing farmland areas and degraded water and soil resources [7,8]. Therefore, for sustainable development, it is critically essential to balance the development between land urbanization and population urbanization.

Urbanization is characterized by the coordinated development of urban and rural areas and urban and rural integration. It requires the transition from disharmony to coordination, preventing aggressive land urbanization and an uncoordinated development of urban areas [9]. Therefore, greater attention should be paid to the important role of people in the urbanization process. Recently, research on urbanization has mainly focused on the coordination degree of land urbanization and population urbanization at the national and provincial levels or in developed areas of China [10]. However, research on the evolution of coordinated relationships at the capacity level is scarce, and little attention has been paid to the coordinated development of capacities, which are special geographical units, in the context of coordinated development of urbanization [11]. It is difficult to accurately characterize the interactive effect between land urbanization and population urbanization because of the nonlinear relationship among the elements in these systems [12]. Therefore, an accurate evaluation of the level of coordination between land and population is effective for assessing the coordinated development of urbanization.

This paper considers Nanchang as an example for improving the coordinated development of land urbanization and population urbanization. Thereby, this study evaluated and discussed the coordinated development of urbanization by establishing an alternative indicator framework. Accordingly, a coupling coordination model was employed to explore and discuss the coordinated development between land urbanization and population urbanization in Nanchang from 2002 to 2017, and the findings of coordination characteristics provide valuable implications for achieving sustainable urban development in the future.

## 2. Literature Review

Recently, the coordinated development of urbanization has been the subject of much research [13]. In this regard, the definition of urbanization focuses on land use intensification, which has been recognized as the most promising form of urban intensification, positively contributing to sustainable cities, mainly because it reduces the pressures of outward expansion. Consequently, many researches have proposed the coordinated development of urbanization from different perspectives [14,15]. In general, land and population are considered two key dimensions for assessing urbanization since coordinated development reflects the relationship between land and population in the urbanization process, which shows that coordinated development also implies human interaction with natural resources [16]. In particular, considering the wide-ranging occurrences in the urbanization process, the

interaction of land and population is one of the urgent national priorities of China. Land urbanization provides a spatial carrier for the increase in the urban population, which indicates that China's land urbanization is closely linked to the large input of the urban population and industrial development. Although the Chinese government has taken actions through the smart growth boundary and policy intervention, it is still struggling to find the best and most effective way to control land urbanization [17]. The boom in population urbanization should accelerate the expansion of urban land, which in turn will lead to an increase in demand for construction land. Particularly, the unprecedented scale of urbanization has also brought serious challenges posed by, for instance, the shift in the concentration of the rural population to urban areas, the extensive use of land, environmental protection, food safety, housing security, the explosion in the urban population and even a series of problems such as "ghost cities" [18]. Hence, elaborate efforts are required to quantify the relationship between land and population in the urbanization process. Consequently, in this paper, the indicators are classified into two dimensions, namely land urbanization and population urbanization.

In fact, many studies on the coordination status of urbanization have usually focused on one or more methods, which have been applied to measure and monitor the extent of the coordinated development of urbanization. For instance, methods of single indicators index, decoupling model and Data Envelopment Analysis (DEA) were employed to analyze the relationship between two elements in the urbanization process [19–21]. However, few studies have discussed the issues related to the land and population in the urbanization process. Fortunately, originating from the field of physics, "coupling" is a concept in which two or more systems influence each other through various interactions [22]. Coupling theory is a principle that describes the relationship between two or more systems that have an impact on each other through an internal mechanism, and also shows that the coupling theory has been applied in various disciplines. In addition, the coupling theory has been proven to be an important tool for working on coordination among two or more systems that have interactions with each other [23]. So it can effectively explain the coordinated development between land urbanization and population urbanization. This study emphasizes synthesizing land urbanization and population urbanization into one dimension since interactions exist among the independent indicators. In particular, one representative interaction is the replacement of farmland by construction land. It demonstrated that two basic forms of value, physical value and economic value, of urbanization are adopted to assess the coordinated development of urbanization [24]. Regrettably, there is few study to analyze the coordinated development relationship in the urbanization process. In general, the primary method of analyzing the characteristics of coordinated development and their influencing factors is based on a single dimension. The two concepts of land urbanization and population urbanization emphasize economic collaboration and the characteristics of coordinated development rather than the influencing factors based on a single dimension. Thus, this paper conducts the coupling coordination model in question to the coordinated development of land and population in the urbanization process.

In referring to the literature introduced above, due to the challenges posed by complexity and data requirements, it is a well-known fact that coordinated development between land and population has become a major topic in urban studies. Many researches have measured coordinated development in China, and also have focused on examining the influencing factors of the dynamic change of rapid urbanization [25]. For instance, per capita disposable income has a potential impact on urbanization. Additionally, the output value of the secondary and tertiary industries as a proportion of GDP can increase the carrying capacity of the labor force [26]. In addition, coordinated development of urbanization, as measured by per capita education expenditures, the amount of public green space and the number of doctors, influence the welfare of the population and are thus related to the coupling degree of urbanization [27]. Meanwhile, research has also found that fixed asset investment continues to rise following shocks from the urban population proportion in urbanization, while the average output value of the secondary and tertiary industries also responds to shocks from the urban population proportion [28]. Besides, land urbanization expansion driven by population urbanization

increases construction land use intensity. The literature also indicates that urbanization changes the economic and social conditions resulting from industrialization [29], and it also shows that the regional disparity caused by urbanization is closely related to changes in land expansion and the population scale [30]. However, much of this work has been concerned with ecological environment protection and urbanization, and few studies have mentioned the fact that the effects of multidimensional variables on urbanization exhibit features of land and other transformations of the population [31]. Thus, there is a necessity to find an appropriate method to facilitate the study in evaluating the coordinated development in the urbanization process. This paper considers the coordinated development as the dependent variable, economic development, industrial structure, public service and location advantages as independent variables, and builds a multivariate linear regression model to study the influence of coordination degree between land and population in the urbanization process [32].

Few studies have considered the changing nature of the coordinated development of urbanization from the perspective of land and population [33]. In particular, there is a lack of studies quantitatively exploring the relationship between land and population in response to urbanization, the effects of socio-economic factors on the status of urbanization, which shows that not only dynamic changes in the characteristics of urbanization but also socio-economic factors affect the coupling degree of urbanization. Therefore, this paper aims to evaluate the coordinated development of urbanization by establishing an alternative indicator framework. This framework can be used to gauge the extent to coordinated development of urbanization. This study used the coupling coordination model (CCM) to assess the transformation between land and population in the urbanization process in the case of Nanchang, China. Furthermore, it analyzes the influencing factors of coordinated development by multivariate linear regression (MLR) model in the urbanization process. Appropriate policies and recommendations are suggested for promoting coordinated development in Nanchang, China.

## 3. Study Area and Methodology

### 3.1. Study Area

Nanchang, which is located in the Poyang Lake Plain, is the capital city of Jiangxi province, China. Jiangxi province borders Fujian, Zhejiang, Anhui, Hubei, Hunan, and Guangdong provinces (Figure 1). Following the implementation of the reform and opening-up policy in China, the Poyang Lake Eco-economic Zone was approved by the State Council in 2009, and it has officially risen to the national strategic level. Meanwhile, as the core city of the Poyang Lake Eco-economic Zone, Nanchang has location, resources and policy advantages. Therefore, Nanchang can obtain benefit from these political advantages and its growing population and construction land.

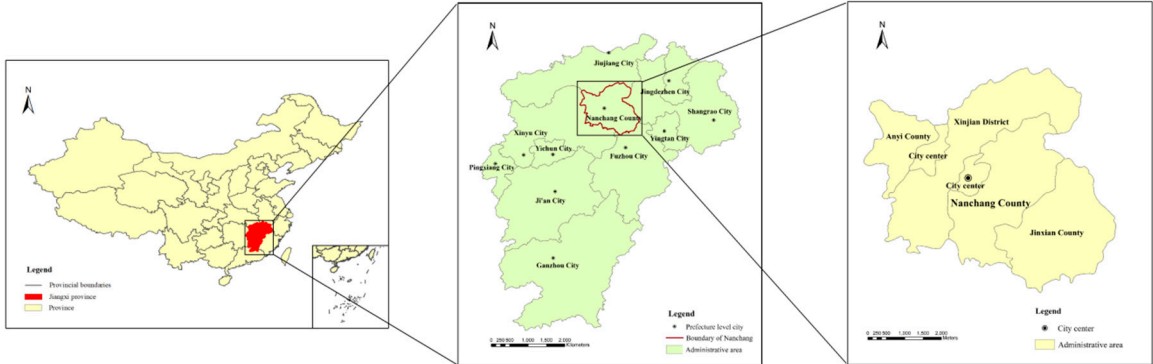

**Figure 1.** Location of Nanchang, Jiangxi province, China.

In addition, during the 2002–2017 period, the registered population of Nanchang increased from 4.42 million to 5.58 million, and the non-agricultural population increased from 1.86 million to 2.36 million [34]. Accordingly, during the past 16 years, the built-up area of Nanchang increased from

87 km² to 327 km², while the per capita disposable income of urban residents increased from 7021 RMB to 20,741 RMB [35]. In particular, all of these statistics suggest a phenomenon of imbalance between land urbanization and population urbanization in Nanchang. Fortunately, regarding the UGB, pilot work on delineating the urban development boundaries in 14 cities across the country has been conducted since 2014 [36]. In this regard, the UGB has been proposed to control the disorderly spread of urban space, and it is a reasonable guide for urban land development and the redevelopment of various natural resources. Meanwhile, in 2017, the policy reform of the household registration system in Jiangxi province was implemented, while this reform clearly cancels the distinction between agricultural hukou system and non-agricultural hukou system, and it establishes a unified urban and rural household registration system. In other words, this reform holds great significance for the orderly promotion of the urbanization of the agricultural transfer population in Nanchang, China. Therefore, with the UGB and household registration system policies that have been implemented in Jiangxi province, it is necessary to evaluate the coordinated development of urbanization from the perspectives of land and population.

*3.2. Framework and Indicators of Urbanization*

Land and population have been considered as two key dimensions to describe urbanization process in previous studies [10]. As two aspects of urbanization, land and population are interdependent. As we know, the land is the basis of population urbanization, which can provide more people with land for housing, education, medical treatment, etc. In addition, it also shows that the increasing population in urban area also poses a threat to the ecological environment of land. Thus, in this context, it is necessary to accommodate a greater urban population by improving the intensive utilization level of various types of land. In fact, population is divided into urban population and rural population; thus, indicators of population urbanization usually include the agricultural population and the non-agricultural population based on household registration (Figure 2) [16]. To avoid the shortcomings of land and population indicators, this paper uses the ratio of urban population to the total population at the end of the year to characterize population urbanization [34]. Therefore, regarding urbanization, population adopts the resident population of the city.

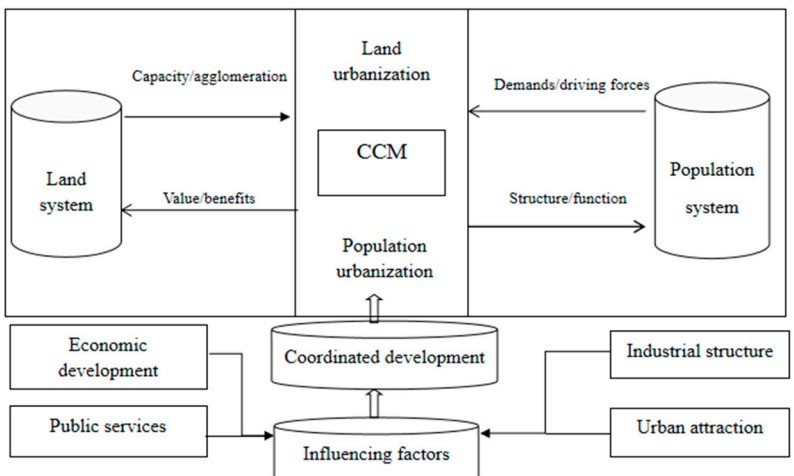

CCM: coupling coordination model.

**Figure 2.** The coordination framework of land and population for urbanization.

In general, as indicators for measuring land urbanization, the construction land area and the built-up area should be considered [37]. Besides, the main difference between them is that the built-up area covers the area of construction land and the area of land acquisition. Therefore, this paper uses the ratio of urban construction land to administrative land area to evaluate land urbanization. We fully

considered the significance of the indicators and the feasibility of their calculations. Therefore, two indicators are considered to measure and characterize the coordinated development of urbanization: land urbanization ($U_1$) and population urbanization ($U_2$) (Table 1).

**Table 1.** Calculation and interpretation of the land-population urbanization level.

| Index | Calculation Method | Indicator Characterization |
|---|---|---|
| Land urbanization ($U_1$) | $U_1 = UCL_i/TL_i$ | The ratio of urban construction land ($UCL_i$) to total land area ($TL_i$) |
| Population urbanization ($U_2$) | $U_2 = PU_i/TP_i$ | The ratio of urban population ($PU_i$) to total population ($TP_i$) |

The main reasons for selecting the CCM are briefly described as follows. First, coupling reflects a process in electron physics that describes the degree of interaction within systems. Meanwhile, the change in the urban population and the expansion of the construction land area are always two basic problems accompanying the urbanization process [38]. In addition, the population is excessively concentrated in the city, which poses increasingly severe challenges to the city's industrial support capacity and basic public service supply capacity. Third, the coupling of coordinated development between land and population combines a variety of factors. In addition, land urbanization is driven by the demand brought by population urbanization and plays an important role in agglomeration and the carrying capacity for the spatial selection of population urbanization [39]. Therefore, a framework for analyzing population and land systems consists of two sub-models. Based on this assessment, the final indicators for coordinated development of urbanization are shown in Figure 2.

*3.3. Assessing Coordinated Development of Urbanization*

3.3.1. Descriptions of Model Construction

This paper focuses on the coordination degree and tests the significant quantitative correlation between land urbanization and population urbanization. In particular, the degree of interaction and mutual influence between the two systems of land and population through their respective factors, which is defined as the degree of coupling and coordination. There are three parts: the classification system and criteria of the development degree (T), coupling degree ($CD_1$), and coordination degree ($CD_2$).

(1) The development degree (T) is calculated by Equation (1). This calculation indicates that the degree of development refers to the evaluation of land urbanization and population urbanization to reflect the coordinated development of urbanization.

$$T = \alpha u_1 + \beta u_2 \tag{1}$$

Here, T represents the development degree, which is a comprehensive evaluation index that reflects the development level of land urbanization and population urbanization, where $U_1$ represents land urbanization, while $U_2$ represents population urbanization. Meanwhile, $\alpha$ and $\beta$ are the weight values to be determined [40]. In addition, considering that the two subsystems of land and population are complementary and equally important, they are given the same weight: $\alpha = \beta = 0.5$.

(2) The coupling degree ($CD_1$) is calculated by Equation (2). It is difficult to reflect the dynamic trend of coordinated development and to make horizontal comparisons because it is easy to commit errors when conducting assessments using single evaluation indices such as the development degree and the coupling degree. Therefore, it is necessary to introduce the coupling degree to measure the relationship between land urbanization and population urbanization. The coupling degree is a quantitative index for measuring coordinated development. In addition, combining the coupling

degree and development degree T reflects the trend of urbanization development from disorder to order.

$$CD_1 = m\left(\frac{(u_1 + u_2 + \cdots u_m)}{(u_1 + u_2 + \cdots u_m)^n}\right)^{1/m} \tag{2}$$

where $CD_1$ represents the coupling degree, and $U_m$ represents the contribution of a certain subsystem m to the total system.

(3) The coordination degree ($CD_2$) is calculated by Equation (3). The degree of coordination expresses the degree of coupling between land and population to assess the interdependence between them.

$$CD_2 = \sqrt{CD_1 \times T} \tag{3}$$

where $CD_2$ represents the coordinated development degree, which integrates the connotation of development degree T and coordination degree $CD_1$. It is an indicator for measuring the overall situation of coordinated development between systems or internal factors. In addition, it has a wide scope of application and can be used for the coordinated development of land urbanization and population urbanization in different regions.

(4) The classification system. Previous studies have suggested that there are ten statuses of coordinated development between two interacting systems, i.e., extreme disorder, serious disorder, moderate disorder, mild disorder, on the verge of disorder, bare coordination, primary coordination, intermediate coordination, favorable coordination and quality coordination [41,42]. The variable $CD_2$ in Equation (3) is a quantitative index that describes the coordination status between land urbanization and population urbanization. In the current study, the classification criteria of $CD_2$ are as Table 2.

**Table 2.** Classification of the degree of coordination between land urbanization and population urbanization.

| Coordination degree | $0.01 < CD_2 \leq 0.10$ | $0.11 < CD_2 \leq 0.20$ | $0.21 < CD_2 \leq 0.30$ | $0.31 < CD_2 \leq 0.40$ | $0.41 < CD_2 \leq 0.50$ |
|---|---|---|---|---|---|
| Coordination level | Extreme disorder | Serious disorder | Moderate disorder | Mild disorder | On the verge of disorder |
| Coordination degree | $0.51 < CD_2 \leq 0.60$ | $0.61 < CD_2 \leq 0.70$ | $0.71 < CD_2 \leq 0.80$ | $0.81 < CD_2 \leq 0.90$ | $0.91 < CD_2 \leq 0.10$ |
| Coordination level | Bare coordination | Primary coordination | Intermediate coordination | Favorable coordination | Quality coordination |
| Contrast type | If $U_1 > U_2$, population lag; if $U_1 < U_2$, land lag | | | | |

### 3.3.2. Statistical Analysis

A multivariate linear regression (MLR) is conducted to analyze the influencing factors of the change in the coupling degree of each research unit. In particular, the variables for the coordination degree are the dependent variables, while 12 indicators are chosen as the independent variables based on the framework in Figure 2. All variables are entered, and stepwise regression is selected to solve the multidisciplinary problem (Table 3). Therefore, the mathematical formula for MLR is expressed as follows:

$$y = \lambda_0 + \lambda_1 x_1 + \lambda_2 x_2 + \lambda_3 x_3 + \ldots + \lambda_{12} x_{12} + \omega \tag{4}$$

where y is the estimated value calculated from the independent variable x; $\lambda_0$ represents the constant term; and $\lambda_1, \lambda_2, \lambda_3, \ldots$, and $\lambda_{12}$ are the partial regression coefficients.

**Table 3.** Variables for assessing the influencing factors of urban sustainability.

| Dimension | Indicators | Definitions | Units |
|---|---|---|---|
| Economic development | Engel coefficient ($x_1$) | Level of economic development | - |
| | Per capita disposable income ($x_2$) | Indicator of the living standards of residents | - |
| | Per capita fixed asset investment ($x_3$) | Investment as a share of GDP | RMB/hm$^2$ |
| Urban industrial structure | Secondary and tertiary industries as a proportion of GDP ($x_4$) | Ratio of the gross product of the secondary and tertiary industries to the regional gross domestic product | RMB/hm$^2$ |
| | Proportion of the labor force in the secondary and tertiary industries ($x_5$) | Ratio of urbanization | % |
| Urban public service | Per capita education expenditures ($x_6$) | Input per unit of education | Persons/km$^2$ |
| | Per capita urban living area ($x_7$) | Per capita area in the city | Persons/km$^2$ |
| | Per capita public green area ($x_8$) | Public green space / total population | Persons/km$^2$ |
| | Number of doctors per 10,000 people ($x_9$) | Ratio of doctors in the city | Persons/km$^2$ |
| Location advantages | Population density ($x_{10}$) | Degree of population agglomeration | Persons/km$^2$ |
| | Urban green coverage rate ($x_{11}$) | Reflects the status of ecological environmental protection in the region | % |
| | Per 10,000 college students ($x_{12}$) | Performance in regional development | % |

### 3.3.3. Data Sources

The data used in this study were mainly sourced from statistical yearbooks. (1) The data for the indicators of land urbanization and population urbanization were collected from the Jiangxi Statistical Yearbook (2002–2017) and China City Statistical Yearbook (2002–2017) [35,36]. (2) The data for variables used to assess the influencing factors were also obtained from the Jiangxi Statistical Yearbook (2002–2017) and China City Statistical Yearbook (2002–2017) [35,36]. (3) Other data were sourced from the website of the Statistical Bureau of Jiangxi.

## 4. Research Results

### 4.1. General Trends of Land Urbanization and Population Urbanization in Nanchang

Figure 3 shows that the proportion of population urbanization grew from 0.421 in 2002 to 0.552 in 2017 at an average rate of 0.82% per year. Meanwhile, the proportion of land urbanization increased from 0.012 in 2002 to 0.044 in 2017 at an average rate of 0.018% per year. In contrast, Figure 3 shows that population urbanization grew more slowly than land urbanization under land finance and industrialization. In this regard, China is at the stage of rapid urbanization, where urban transformation is dependent on a continuous supply of land.

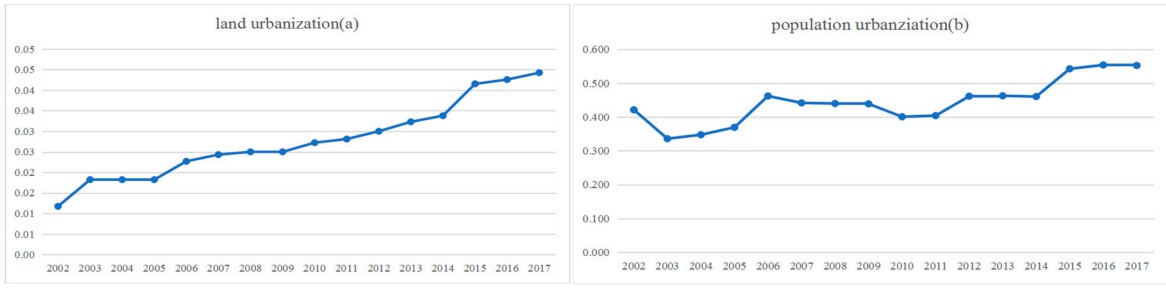

**Figure 3.** Temporal trend of land urbanization (**a**) and population urbanization (**b**) in Nanchang from 2002 to 2017.

### 4.2. Development Degree and Coupling Degree Analysis

As mentioned above, the temporal changes in urbanization in Nanchang from 2002 to 2017 are shown in Table 4. Regarding the indicators of the coordinated development of urbanization, the development degree (*T*) presents an increasing trend. In addition, the level of development of urbanization in Nanchang increased from 0.216 to 0.298 during this 16-year period, with an average annual increase of 0.018. Although the level of increase in the development degree is not high, it also reflects that the overall level of urbanization in Nanchang is better and that urban construction, with population and land as the core, promotes the overall development of Nanchang.

**Table 4.** The dynamic relationship of urbanization from 2002 to 2017.

| Year | LU | PU | T | $CD_1$ | $CD_2$ | Coordination Level | Contrast Type |
|------|------|------|------|------|------|------|------|
| 2002 | 0.012 | 0.421 | 0.216 | 0.151 | 0.181 | Serious disorder | Land lag |
| 2003 | 0.018 | 0.336 | 0.177 | 0.186 | 0.181 | Serious disorder | Land lag |
| 2004 | 0.018 | 0.348 | 0.183 | 0.186 | 0.184 | Serious disorder | Land lag |
| 2005 | 0.018 | 0.369 | 0.194 | 0.186 | 0.190 | Serious disorder | Land lag |
| 2006 | 0.023 | 0.462 | 0.242 | 0.208 | 0.225 | Moderate disorder | Land lag |
| 2007 | 0.024 | 0.442 | 0.233 | 0.215 | 0.224 | Moderate disorder | Land lag |
| 2008 | 0.025 | 0.440 | 0.232 | 0.217 | 0.225 | Moderate disorder | Land lag |
| 2009 | 0.025 | 0.439 | 0.232 | 0.217 | 0.225 | Moderate disorder | Land lag |
| 2010 | 0.027 | 0.401 | 0.214 | 0.226 | 0.220 | Moderate disorder | Land lag |
| 2011 | 0.028 | 0.404 | 0.216 | 0.229 | 0.223 | Moderate disorder | Land lag |
| 2012 | 0.030 | 0.461 | 0.246 | 0.237 | 0.241 | Moderate disorder | Land lag |
| 2013 | 0.032 | 0.463 | 0.247 | 0.246 | 0.247 | Moderate disorder | Land lag |
| 2014 | 0.034 | 0.460 | 0.247 | 0.251 | 0.249 | Moderate disorder | Land lag |
| 2015 | 0.042 | 0.542 | 0.292 | 0.278 | 0.285 | Moderate disorder | Land lag |
| 2016 | 0.043 | 0.554 | 0.298 | 0.281 | 0.290 | Moderate disorder | Land lag |
| 2017 | 0.044 | 0.552 | 0.298 | 0.286 | 0.292 | Moderate disorder | Land lag |

The coupling degree ($CD_1$) results show an increasing trend from 2002 to 2017 in Nanchang. In addition, the results indicate that the coupling degree is an index reflecting the degree of internal dependence between land and population. Furthermore, the degree of coupling between land urbanization and population urbanization in Nanchang increased from 0.151 in 2002 to 0.286 in 2018, with an average annual increase of 0.031. The interdependence between land and population increases year by year in the process of urbanization, which also means that urban population agglomeration promotes land urbanization; that is, the increase in population requires more urban residential, educational, medical and public green space to meet the basic quality of life requirements of residents. On the other hand, land urbanization promotes population urbanization. By improving the living environment and optimizing the pattern of industrial land, more people will be attracted to urban employment and residence.

The coordination degree ($CD_2$) results demonstrate an increasing trend between 2002 and 2017 in Nanchang. The process can be divided into three phases: (1) From 2002 to 2004, the coordination degree indicates that the growth rate of the population urbanization index was slightly higher than that of the land urbanization index. (2) From 2005 to 2009, the growth rate of the land urbanization index was still lower than that of the population urbanization index, and the two indices were relatively close and relatively stable. (3) From 2010 to 2017, the land urbanization index in this stage increased substantially. Since 2008, with large-scale national macro-control investment, the scale effect of urban land in Nanchang has gradually emerged. The urban land of the city increased from 87 $km^2$ in 2002 to 327 $km^2$ in 2017, with an average annual growth of 7%, while the urban household registration population increased from 1.86 million to 2.36 million, with an average annual growth of 3.12%. Therefore, these results suggest that the average annual growth rate of land urbanization is higher than that of population urbanization, which to some extent proves that at present, land urbanization is taking place faster than population urbanization in Nanchang.

Table 4 presents the contrast analysis of urbanization. In 2002–2005, the level of land urbanization lagged behind the level of population urbanization, showing a land lag; in 2004–2017, the level of population urbanization lagged behind the level of land urbanization, showing a population lag,

from the lag of population urbanization to the lag of land urbanization. The reason is that the local government could obtain a high income from land transfer proceeds. In return, the government could guarantee a large-scale supply of urban land to the market by expropriating the land around the city, especially in the Poyang Lake Eco-economic Zone, which rose to the national strategy level in 2009. As the core area of urban development, Nanchang entered the land market with a large amount of social capital, causing the level of land urbanization to peak in 2010. However, due to the restriction of the household registration system, it was difficult for the floating population to settle down, which eventually led to the divergence between land urbanization and population urbanization.

### 4.3. The Division of the Coordination Stages of Urbanization in Nanchang

According to the coordination degree, the urbanization process can be divided into the serious disorder stage and the moderate disorder stage.

(1) Serious disorder stage (2002–2005). The coordination degree increased from 0.181 to 0.190, reflecting that the gradual change in the land and population systems was still in a low recession stage. In addition, the results show that the land urbanization index and population urbanization index of Nanchang were gradually converging and that the corresponding coordination degree was still in a low recession stage from 2002 to 2005.

(2) Moderate disorder stage (2006–2017). The coordination degree increased from 0.225 to 0.292, and the average coordination level was 0.245. The degree of coordination between land urbanization and population urbanization was more orderly but remained low.

Figure 4 shows the dynamic degree of coordination between the two processes involved over the 2002–2017 period. The coordination degree presented an increasing trend but remained at a low level. As mentioned above, there was a lack of synchronous development of the processes involved. In fact, the overall trend of coordination degree increased from 2002 to 2017, but slightly slowed down after 2006. In other words, although land urbanization underwent substantial advancement, its increasing pace still lagged behind that of population urbanization. Uncoordinated urban-rural development and urban-rural inequalities in income have increased in a greater number of urbanizing places of China. Thus, in 2008, urbanization became a key national development strategy [31]. Therefore, to pursue political promotion, policy makers have been eager to promote urbanization to achieve economic growth. In addition, new construction and projects were authorized, ignoring the actual demand for land development. Furthermore, through administrative adjustment, the former rural population in suburban areas became urban population.

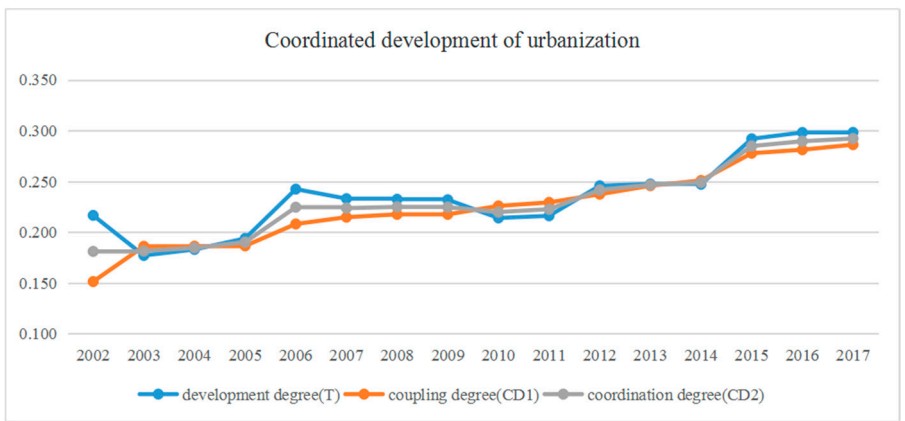

**Figure 4.** Dynamic coordination degree of urbanization in Nanchang from 2002 to 2017.

## 5. Discussion and Policy Recommendations

*5.1. Discussion*

5.1.1. Characteristics of the Coordinated Development of Urbanization

The coordinated development of the changes in urbanization in Nanchang has various characteristics. The degree of coordination between land urbanization and population urbanization in Nanchang has increased since 2002 [43]. This increase quickly raises the scale of urban land in the urbanization process. However, the increasing trend of population urbanization changed in early 2015. Nanchang, which is in a developing province in central China, has urgently raised the rate of urbanization; however, coordination degree of urbanization has emerged in the past 16 years. Furthermore, one important reason for this phenomenon is the fact that the population urbanization of Nanchang lags behind that of other capital cities, which is recognized as a major future challenge in most cities in China.

In contrast, land urbanization has maintained a steady increase in Nanchang since 2002. The increasing land urbanization denotes that economic benefits are the main objective in Nanchang, suggesting that the capital city is characterized by high-intensity construction land use in the urbanization process. However, the results reveal that land urbanization lags behind population urbanization. In other words, the intensive construction land use in Nanchang is not paying sufficient attention to boosting public services and the urban industrial structure or to the city's location advantages in the process of rapid economic growth. Besides, the change trends have a synchronous tempo relationship between land and population. These characteristics of the changes in land urbanization correspond to the entirely different influencing factors of economic transition. Thus, this implies that public services are not achieved by simultaneously increasing the land and population in Nanchang. More varied influencing factors of coordinated development of urbanization affect these change trends in the coordination degree.

Notably, in Nanchang, $CD_2$ increased from 2002 to 2017, while $CD_2$ fluctuated or decreased after 2011. On the one hand, due to the beginning of the implementation of the UGB and the household registration system policy reform, meant to boost population urbanization, we can infer that due to its stringency, land management improved remarkably around the first year of the policy. In addition, the improved coordination status of urbanization demonstrated the effectiveness of the policy. However, over the next few years, the coordination degree of Nanchang failed to maintain a coordination status at a high level or did not show sustainable improvement, suggesting that policy implementation is complex and faces the challenge of uncertainty. Therefore, to make a long-term difference, the policies should be strictly adhered to. For instance, close collaboration between land and population, as well as sufficient safeguard measures, is needed to guarantee the persistent, effective implementation of the UGB and household registration system policy reform meant to boost population urbanization.

5.1.2. Influencing Factors Associated with Urbanization in Nanchang

Table 5 presents the influencing factors associated with urbanization in Nanchang. The results reveal that the MLR models are suitable for predicting the response of land urbanization and population urbanization since over 89% of the likelihood ratio (LR) has been explained. Therefore, this model can be used to explain the influencing factors of the coordination degree.

**Table 5.** The influencing factors of the coordination degree of urbanization.

| Dimension | Variables | y = T | y = CD$_1$ | y = CD$_2$ |
|---|---|---|---|---|
| Economic development | x$_1$ | 0.0426 * (0.0158) | 0.0373 * (0.0170) | 0.0407 ** (0.0144) |
| | x$_2$ | 0.516 *** (0.0547) | 0.229 ** (0.0587) | 0.373 *** (0.0499) |
| | x$_3$ | 0.274 *** (0.0388) | 0.142 ** (0.0416) | 0.208 *** (0.0354) |
| Urban industrial structure | x$_4$ | 0.209 *** (0.0212) | 0.167 *** (0.0227) | 0.190 *** (0.0193) |
| | x$_5$ | 0.104 *** (0.0131) | 0.0367 * (0.0140) | 0.0687 *** (0.0119) |
| Urban public services | x$_6$ | −0.774 *** (0.0661) | −0.358 *** (0.0709) | −0.564 *** (0.0602) |
| | x$_7$ | 0.0212 (0.0158) | 0.0741 ** (0.0169) | 0.0478 ** (0.0144) |
| | x$_8$ | −0.145 *** (0.0100) | −0.0541 *** (0.0108) | −0.0985 *** (0.00915) |
| | x$_9$ | 0.00127 (0.0172) | −0.00497 (0.0184) | −0.00439 (0.0156) |
| Location advantage | x$_{10}$ | 0.0246 (0.0139) | −0.0125 (0.0149) | 0.00408 (0.0127) |
| | x$_{11}$ | 0.145 *** (0.0214) | 0.0800 ** (0.0230) | 0.113 *** (0.0195) |
| | x$_{12}$ | −0.159 *** (0.0216) | −0.0923 ** (0.0232) | −0.124 *** (0.0197) |
| Constant | | 0.131*** (0.0158) | 0.104 *** (0.0170) | 0.116 *** (0.0144) |
| LR | | 89.71 | 88.43 | 92.34 |
| Observations | | 16 | 16 | 16 |

Note: ***, **, and * indicate significance at the 1%, 5%, and 10% levels, respectively. Method: stepwise regression. Abbreviations: development degree (T); coupling degree (CD$_1$); coordination degree (CD$_2$).

The variables that describe urbanization have multiple effects on the coordinate degree. The influencing factors of per capita education expenditures (x$_6$), the per capita public green area (x$_8$), and per 10,000 college students (x$_{12}$) have negative effects on the coordination degree. Therefore, these results mean that increasing per capita education expenditures (x$_6$), the per capita public green area (x$_8$), and per 10,000 college students (x$_{12}$) will influence the coordination degree in Nanchang.

Meanwhile, the influencing factors of per capita disposable income (x$_2$), per capita GDP (x$_3$), the proportion of the secondary and tertiary industries in GDP (x$_4$), per capita fixed asset investment (x$_5$), per capita education expenditures (x$_7$), population density (x$_{10}$), and the urban green coverage rate (x$_{11}$) have positive effects on the coordination degree. All of these results suggest that population urbanization can increase the demand for construction land. Hence, economic development, urban industrial structure, urban public services and location advantages can increase the regression coefficient for coordinated development of urbanization.

Generally, regarding the economic development dimension, the influencing factors of land urbanization and population urbanization with regard to the coupling degree from 2002 to 2017 are analyzed using MLR. From the regression results, per capita disposable income (x$_2$) and per capita GDP (x$_3$) are positively correlated with the coordination degree of urbanization in 2017; the regression coefficients are 0.373 and 0.208, respectively. Regarding the urban industrial structure dimension, the proportion of the secondary and tertiary industries in GDP (x$_4$) and per capita fixed asset investment (x$_5$) are positively correlated with the coordination degree of population urbanization and land

urbanization, with regression coefficients of 0.190 and 0.069, respectively. Regarding the urban public services, the factor of per capita education expenditures ($x_7$) has a positive effect on the coordination degree; the regression coefficient is 0.487. Furthermore, regarding the location advantage dimension, the influencing factor of the urban green coverage rate has a positive effect on the coordination degree, with a regression coefficients of 0.113.

As mentioned above, the degree of concentration of urban population during this period is the major factor promoting the coupling pattern of land urbanization and population urbanization in Nanchang, and the coordination degree of the influencing factors shows an increasing trend. In particular, this result suggests that the degree of concentration of urban population affects the population urbanization and promotes the industrial structure. Thus, the coupling pattern of land urbanization and population urbanization is affected. Economic development, urban industrial structure, urban public services and location advantages also have an important impact on the coupling pattern from 2002 to 2017. However, the influence of per 10,000 college students is weakened, which means that the effect of the efficiency of urban expansion begins to appear. Therefore, improving public services through efficiency in urban expansion provides the main impetus for the coordinated development of urbanization.

### 5.1.3. Hidden Causes of Changes in the Coordinated Development of Urbanization

Land and population play an important role in the temporal trend of urban sustainability. The MLRs provide some important quantitative insights regarding the changes in land urbanization in response to population urbanization in Nanchang. There are hidden causes of the effects of population urbanization on land urbanization.

In fact, the constraints imposed by external factors account for this situation. Based on the contrast analysis, Nanchang still shows a land lag, and the coordination degree has always been in place, which is related to external institutional constraints. The most important aspect is the influence of different institutional arrangements of land expropriation and the land transfer market. On the one hand, based on the non-marketization of land expropriation and the marketization of land transfer, a dual income gap has been formed, which has greatly promoted the development of land urbanization in Nanchang. On the other hand, due to the different price strategies of industrial land and residential land transfer, the land transfer prices of industrial land and residential land in Nanchang in 2011 were 4.90 million RMB/hm$^2$ and 60.26 million RMB/hm$^2$, respectively. However, the high land price strategy of residential land raised the price of residential land, increased the cost of migration, and ultimately inhibited population urbanization [34]. Furthermore, the urban-rural dual household registration conversion costs has an impact. This factor is closely related to population urbanization, which needs to bear the costs of education, medical treatment, employment and public infrastructure. Presently, the per capita public cost of urbanization of the agricultural transfer population in China is approximately 130,000 RMB [35,36]. The huge public financial expenditure has restrained population urbanization in Nanchang.

Meanwhile, the results show the response to the constraints imposed by internal factors. Under the optimal status of urbanization, the simultaneous development of land urbanization and population urbanization is achieved. During the study period, the development degree of urbanization in Nanchang was lower than the coupling degree of urbanization, which is closely related to the overall low level of urban construction in Nanchang. This is a result of the weaker influence of the strength of social-economic development. Because Nanchang is located in central China, it lacks the relative location advantages of the coastal areas, which is reflected in the relatively lag in the comprehensive strength of urbanization, investment density, the industrial structure and openness, and capital [44–46], as well as the impact of inadequate investment in technology and other supporting infrastructure. Despite the rapid development of industrialization in Nanchang in the past 16 years, the district of the Honggutan Area and High-Tech Industrial Development Zone has been planned and developed. However, behind the large-scale expansion of urban construction land, the "absorptive effect" of

investment in supporting infrastructure such as per capita public green space, doctors per capita and per capita fixed asset investment has been insufficient, and internal factors show a relatively low level of coordination [47].

*5.2. Policy Recommendations*

The imbalance of land urbanization and population urbanization worsens the relationship between them and will not allow sustainable development in mega-cities to be maintained [48]. In particular, the key factors are the dual pricing strategy and the fixed household registration system, while the internal factors are constrained by the relatively lagging social and economic development. Clearly, the coordination level has changed from serious disorder to moderate disorder. Therefore, we should optimize the forms of urbanization to balance land urbanization. First, it is necessary to promote collective construction land in the market, implement diversified urban land supply channels and strengthen the land market [49]. In addition, the gap between urban and rural areas should be narrowed. As one important measure, a balanced land price mechanism should be built. Furthermore, to restrict the scale of new urban development zones in Nanchang, the boundaries of new urban construction and development zones should be delimited to realize the orderly expansion of the urban scale and to promote the redevelopment of the urban land stock. Furthermore, it is necessary to supervise urban land transfer funds, and more attention should be paid to incorporating land transfer funds into the revenue of the fiscal budget, constructing a public financial management framework with land transfer funds as the core, and solving the problem posed by the excessive dependence on land finance in urban development to avoid excessive land urbanization.

## 6. Conclusions

A CCM of development for urbanization was built based on land and population indicators, and the yearly $CD_2$ of urban sustainability for Nanchang, China, was calculated during the 2002–2017 period. Then, the impacts of the coordination degree on urbanization were quantified using MLR. Furthermore, some relevant policy recommendations were given to help develop a coordinated and balanced land urbanization and population urbanization in Nanchang. The following important conclusions are described as follows:

(1) The coordinated development of urbanization at the city scale in Nanchang fluctuated during the 2002–2017 period and increased significantly after the implementation of the UGB policy and the preferential policies for attracting talents. The average $CD_2$ of urban sustainability in Nanchang is greater than ever before.

(2) The finding shows that the relationship between land urbanization and population urbanization in Nanchang is inconsistent, which means that the relationship is always in a stage of non-synchronous development. Although the coordination degree of land urbanization and population urbanization in Nanchang has gradually become coordinated, the coordination level still shows a status of moderate disorder.

(3) For Nanchang, the degree of coordination of land and population has improved, and economic growth was rapid during the 2002–2017 period, and all of these factors play an important role in the change in the provincial coordination degree, mainly in the form of increasing the land supply, demand and pollution. However, the public green area has an adverse impact on the coordination degree of urban sustainability in the post-phase.

(4) The implementation of the UGB has not only improved the coordination status between land and population but has also decreased the disparity of the coordination degree of urban sustainability in Nanchang. Therefore, the UGB should be effectively implemented and adhered to for the policy to make a lasting difference. Moreover, cross-regional cooperation and communication should be advocated by the local government to achieve balanced and coordinated development of urbanization in the long term.

**Author Contributions:** Conceptualization, T.L. and H.X.; methodology, T.L. and W.L.; software, T.L. and L.W.; validation, T.L. and W.L.; formal analysis, H.L. (Hua Lu); investigation, Y.Z.; resources, T.L.; data curation, T.L.; writing—original draft preparation, T.L. and X.Z.; writing—review and editing, T.L. and X.Z.; visualization, H.L. (Hongyi Li); supervision, H.X.; project administration, T.L.; funding acquisition, T.L.

**Funding:** This study was supported by the National Natural Science Foundation of China (No. 71864016), the China Postdoctoral Science Foundation (No.2017M622098), the Postdoctoral Science Foundation of Jiangxi Province(No.2017KY55); the Postdoctoral for Daily Funding of Jiangxi Province (No. 2017RC036), the Humanity and Social Science Youth Foundation of the Ministry of Education of China (No. 17YJC630100), the Natural Science Foundation of Jiangxi Province (No.20171BAA218017), the Humanities and Social Sciences Fund Project of Jiangxi Province (No.GL18242), the "13th Five-Year" planning project of Jiangxi Academy of Social Science (No.16GL31) and the Technology Foundation of Jiangxi Education Department (No. GJJ160460).

**Conflicts of Interest:** The authors have no conflicts of interest to declare.

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
