# Peer review of "Coupling Coordinated Development and Exploring Its Influencing Factors in Nanchang, China: From the Perspectives of Land Urbanization and Population Urbanization"

_land, doi:10.3390/land8120178_

Round 1

Reviewer 1 Report

This manuscript is acceptable in the current form.

Author Response

Response to Reviewer 1

Point 1. This manuscript is acceptable in the current form.

Response 1: Thank you very much for accepting our manuscript (642709), entitled “Coupling Coordinated Development and Exploring Its Influencing Factors in Nanchang, China: From the Perspectives of Land Urbanization and Population Urbanization”.

Reviewer 2 Report

The paper "Coupling the Coordinated Development and Exploring Its Influential Factors from the Perspectives of Land Urbanization and Population Urbanization: A Case of Nanchang, China" empirically analyses the interaction between land urbanization and population urbanization in Nanchang from 2002 to 2017 based on the coupling model.
The paper is well written (I would like to suggest a professional peer review service) and it is well organised.

I would like to stress a few points to take into consideration during the submission of the revised paper:
1. The title is totally unclear. It is too long and it is not able to summarise the contents of the paper.
2. The coupling model is not well explained in the paper. Please consider adding a paragraph on it.
3. I suggest the authors expand the references referring in addition to a more worldwide perspective.

Author Response

Response to Reviewer 2

 Point 1.The paper "Coupling the Coordinated Development and Exploring Its Influential Factors from the Perspectives of Land Urbanization and Population Urbanization: A Case of Nanchang, China" empirically analyses the interaction between land urbanization and population urbanization in Nanchang from 2002 to 2017 based on the coupling model. The paper is well written (I would like to suggest a professional peer review service) and it is well organised.

Response 1: Thank you very much for the opportunity to allow us to improve our manuscript. We took reviewer’s suggestion. We have revised our manuscript thoroughly by addressing each and every comment by the English editing service organization. We have also checked the English of the manuscript throughout to eliminate grammatical errors.

Point 2.The title is totally unclear. It is too long and it is not able to summarise the contents of the paper.

Response 2: We took reviewer’s suggestion and modified the title in the paper, which made it more clear than before.

Line1-Line5: “Coupling Coordinated Development and Exploring Its Influencing Factors in Nanchang, China: From the Perspectives of Land Urbanization and Population Urbanization”.

Point 3.The coupling model is not well explained in the paper. Please consider adding a paragraph on it.

Response 3: We took reviewer’s suggestion and added more explains with regard to CCM in this paper.

Line 244-Line 256: The main reasons for selecting the CCM are briefly described as follows. First, coupling reflects a process in electron physics that describes the degree of interaction within systems. Meanwhile, the change in the urban population and the expansion of the construction land area are always two basic problems accompanying the urbanization process [38]. In addition, the population is excessively concentrated in the city, which poses increasingly severe challenges to the city's industrial support capacity and basic public service supply capacity. Third, the coupling of coordinated development between land and population combines a variety of factors. In addition, land urbanization is driven by the demand brought by population urbanization and plays an important role in agglomeration and the carrying capacity for the spatial selection of population urbanization [39]. Therefore, a framework for analysing population and land systems consists of two sub-models. Based on this assessment, the final indicators for coordinated development of urbanization are shown in Fig. 2.

Point 4. I suggest the authors expand the references referring in addition to a more worldwide perspective.

Response 4: According to reviewer’s suggestion, we added some references across global perspective.

Line 662

23.Vojnovic, I. Urban sustainability: Research, politics, policy and practice. Cities,2014, 41, 30–44.

Line696-Line 697

Manoj, R. Planning for sustainable urbanization in fast growing cities: Mitigation and adaptation issues addressed in Dhaka, Bangladesh. Habitat International,2009,33,276–286

Line 700

40.Alexander,C. Micropolitan areas and urbanization processes in the US.Cities,2012,29,24-28

Line 712-Line 716

Markus, B. Adult mortality and urbanization: examination of a weak connection in sub-Saharan Africa. World Development,2019,122,184-198. Juliane,D.; Susanne,S.; Judith, M.; Marcus, N. Urbanization and socio-ecological challenges in high mountain towns: Insights from Leh (Ladakh), India. Landscape and Urban Planning,2019,189,189-199.

Reviewer 3 Report

First, I would like to congratulate the authors for the paper "Coupling the coordinated development and exploring its influential factors from the perspectives of land urbanization and population urbanization: A case of Nanchang, China".
The structure of the article is practically perfect, highlighting the fact of adding policy recommendations in the discussion section.
The Introduction and Literature review sections are very complete, but I would recommend the following points to be taken into account and improve the study:
- Adding the description of the paper structure at the end of the introductory section.
- Adding some reference to the methodology used and the reasons for using it in the Introduction.
- Adding bibliographic citations about similar studies in other regions, whether Asian or from other continents.
The methodology is well justified, but I think that citing other papers where it is also used would be adequate. For example:
Ma, H., & Du, J. (2012). Influence of Industrialization and Urbanization on China’s Energy Consumption. In Advanced Materials Research (Vol. 524, pp. 3122-3128). Trans Tech Publications

Author Response

Response to Reviewer 3

 Point 1: First, I would like to congratulate the authors for the paper "Coupling the coordinated development and exploring its influential factors from the perspectives of land urbanization and population urbanization: A case of Nanchang, China".

Response 1: Thank you very much for the opportunity to allow us to improve our manuscript.

Point 2: The structure of the article is practically perfect, highlighting the fact of adding policy recommendations in the discussion section.

Response: Thank you very much for the opportunity to allow us to improve our manuscript. It must be noted that the policy recommendations would be benefit to achieve the goal of sustainable development in Nanchang.

Line 554-Line 572: The imbalance of land urbanization and population urbanization worsens the relationship between them and will not allow sustainable development in mega-cities to be maintained [48]. In particular, the key factors are the dual pricing strategy and the fixed household registration system, while the internal factors are constrained by the relatively lagging social and economic development. Clearly, the coordination level has changed from serious disorder to moderate disorder. Therefore, we should optimize the forms of urbanization to balance land urbanization. First, it is necessary to promote collective construction land in the market, implement diversified urban land supply channels and strengthen the land market [49]. In addition, the gap between urban and rural areas should be narrowed. As one important measure, a balanced land price mechanism should be built. Furthermore, to restrict the scale of new urban development zones in Nanchang, the boundaries of new urban construction and development zones should be delimited to realize the orderly expansion of the urban scale and to promote the redevelopment of the urban land stock. Furthermore, it is necessary to supervise urban land transfer funds, and more attention should be paid to incorporating land transfer funds into the revenue of the fiscal budget, constructing a public financial management framework with land transfer funds as the core, and solving the problem posed by the excessive dependence on land finance in urban development to avoid excessive land urbanization.

Point 3: Adding the description of the paper structure at the end of the introductory section.
Response 3: We took reviewer’s suggestion and modified the structure in the paper.

Line 82-Line 89: This paper considers Nanchang city as an example for improving the coordinated development of land urbanization and population urbanization. Thereby, this study evaluated and discussed the coordinated development of urbanization by establishing an alternative indicator framework. Accordingly, a coupling coordination model was employed to explore and discuss the coordinated development between land urbanization and population urbanization in Nanchang from 2002 to 2017, and the findings of coordination characteristics provide valuable implications for achieving sustainable urban development in the future.

Point 4: Adding some reference to the methodology used and the reasons for using it in the Introduction.

Response 4: We took the reviewer’s advices, present the reference to the methodology used in the introduction. (1) the reasons for using the methodology are described as follows:

Line123-Line 129: Fortunately, originating from the field of physics, “coupling” is a concept in which two or more systems influence each other through various interactions [22]. Coupling theory is a principle that describes the relationship between two or more systems that have an impact on each other through an internal mechanism, and also shows that the coupling theory has been applied in various disciplines. In addition, the coupling theory has been proven to be an important tool for working on coordination among two or more systems that have interactions with each other [23].

(2)The references of methodology are added as follows:

Line 653-Line 659:

You, H. Quantifying the coordinated degree of urbanization in Shanghai, China. Quality & Quantity, 2016, 3,1273-1283. Liu,H.; Liu,Y.; Wang,H.; Yang,J.; Zhou, X. Research on the coordinated development of greenization and urbanization based on system dynamics and data envelopment analysis—A case study of Tianjin. Journal of Cleaner Production, 2019,214, 195-208. Peng, X.Urban land use efficiency changes and improvement strategies from a decoupling perspective in Nanchang City. Resources Science,2016,3,493-500 (in Chinese)

Point 5: Adding bibliographic citations about similar studies in other regions, whether Asian or from other continents.

Response 5: According to reviewer’s suggestion, we added the references at the end of this sentences.

Line 662

23.Vojnovic, I. Urban sustainability: Research, politics, policy and practice. Cities,2014, 41, 30–44.

Line696-Line 697

Manoj, R. Planning for sustainable urbanization in fast growing cities: Mitigation and adaptation issues addressed in Dhaka, Bangladesh. Habitat International,2009,33,276–286

Line 700 -Line 702

40.Alexander,C. Micropolitan areas and urbanization processes in the US.Cities,2012,29,24-28

41.Pramit, V.; Raghubanshi. Urban sustainability indicators: Challenges and opportunities. Ecological Indicators,2018,93,282–291.

Line 712-Line 716

Markus, B. Adult mortality and urbanization: examination of a weak connection in sub-Saharan Africa. World Development,2019,122,184-198. Juliane,D.; Susanne,S.; Judith, M.; Marcus, N. Urbanization and socio-ecological challenges in high mountain towns: Insights from Leh (Ladakh), India. Landscape and Urban Planning,2019,189,189-199.

Point 6: The methodology is well justified, but I think that citing other papers where it is also used would be adequate. For example: Ma, H., & Du, J. (2012). Influence of Industrialization and Urbanization on China’s Energy Consumption. In Advanced Materials Research (Vol. 524, pp. 3122-3128). Trans Tech Publications

 Response 6: According to reviewer’s suggestion, we added the references at the end of this sentences.

Line689-Line 690:

Ma, H.; Du, J. Influence of industrialization and urbanization on China’s energy consumption. Advanced Materials Research, 2012, 524:3122-3128.
